# Samba: Simple Hybrid State Space Models for Efficient Unlimited Context Language Modeling

**Liliang Ren**[1,2*] **Yang Liu**[1†] **Yadong Lu**[1†] **Yelong Shen**[1] **Chen Liang**[1] **Weizhu Chen**[1]

[1]Microsoft    [2]University of Illinois at Urbana-Champaign

{liliangren,yaliu10,yadonglu,yelong.shen,chenliang1,wzchen}@microsoft.com

## Abstract

Efficiently modeling sequences with infinite context length has long been a challenging problem. Previous approaches have either suffered from quadratic computational complexity or limited extrapolation ability in length generalization. In this work, we present Samba, a simple hybrid architecture that layer-wise combines Mamba, a selective State Space Model (SSM), with Sliding Window Attention (SWA). Samba selectively compresses a given sequence into recurrent hidden states while still maintaining the ability to precisely recall recent memories with the attention mechanism. We scale Samba up to 3.8B parameters with 3.2T training tokens and demonstrate that it significantly outperforms state-of-the-art models across a variety of benchmarks. Pretrained on sequences of 4K length, Samba shows improved perplexity in context lengths of up to 1M in zero-shot. When finetuned on 4K-length sequences, Samba efficiently extrapolates to a 256K context length with perfect memory recall on the Passkey Retrieval task, and exhibits superior retrieval extrapolation on the challenging Phonebook task compared to full-attention models. As a linear-time sequence model, Samba achieves a $3.73\times$ higher throughput compared to Transformers with grouped-query attention for user prompts of 128K length, and a $3.64\times$ speedup when generating 64K tokens with unlimited streaming. Our code for training on open source data is publicly available at https://github.com/microsoft/Samba.

## 1 Introduction

Attention-based models (Vaswani et al., 2017; Bahdanau et al., 2014) have dominated the neural architectures of Large Language Models (LLMs) (Radford et al., 2019; Brown et al., 2020; OpenAI, 2023; Bubeck et al., 2023) due to their ability to capture complex long-term dependencies and the efficient parallelization for large-scale training (Dao et al., 2022a). Recently, State Space Models (SSMs) (Gu et al., 2021; Smith et al., 2023; Gu et al., 2022; Gu & Dao, 2023) have emerged as a promising alternative, offering linear computation complexity and the potential for better extrapolation to longer sequences than seen during training. Specifically, Mamba (Gu & Dao, 2023), a variant of SSMs equipped with selective state spaces, has demonstrated notable promise through strong empirical performance and efficient hardware-aware implementation. Recent work also shows that transformers have poorer modeling capacities than input-dependent SSMs in state tracking problems (Merrill et al., 2024). However, SSMs struggle with memory recall due to their recurrent nature (Arora et al., 2023), and experimental results on information retrieval-related tasks (Fu et al., 2023; Wen et al., 2024; Arora et al., 2024), have further shown that SSMs are not as competitive as their attention-based counterparts.

Previous works (Zuo et al., 2022; Fu et al., 2023; Ma et al., 2023; Ren et al., 2023) have explored various approaches to hybridize SSMs with the attention mechanism, but none have demonstrated significantly better language modeling performance compared to state-of-the-art Transformer architectures. Existing length extrapolation techniques (Han et al., 2023; Xiao et al., 2023; Jin et al.,

---

*Work partially done during internship at Microsoft.

†Equal second-author contribution.

2024) designed for attention mechanisms are constrained by quadratic computational complexity or insufficient context extrapolation performance, particularly when evaluated under perplexity metrics. In this paper, we introduce SAMBA, a simple neural architecture that harmonizes the strengths of both the SSM and the attention-based models, while achieving a potentially infinite length extrapolation with linear time complexity. SAMBA combines SSMs with attention through layer-wise interleaving Mamba (Gu & Dao, 2023), SwiGLU (Shazeer, 2020), and Sliding Window Attention (SWA) (Beltagy et al., 2020). Mamba layers capture the time-dependent semantics and provide a backbone for efficient decoding, while SWA fills in the gap modeling complex, non-recurrent dependencies. A detailed discussion of related work is included in Appendix A.

We scale SAMBA with 421M, 1.3B, 1.7B and up to 3.8B parameters with 3.2T tokens. In particular, the largest 3.8B post-trained model achieves a 71.9 score for MMLU (Hendrycks et al., 2021), 62.8 for HumanEval (Chen et al., 2021), and 87.6 for GSM8K (Cobbe et al., 2021), substantially outperforming the post-trained Phi-3-mini model under a control of the same training recipes and datasets, as detailed in Table 1. Despite being pre-trained in the 4K sequence length, SAMBA can be extrapolated to 1M length in zero shot with improved perplexity on Proof-Pile (Zhangir Azerbayev & Piotrowski, 2022), achieving a 256× extrapolation ratio, while still maintaining the linear decoding time complexity with unlimited token streaming, as shown in Figure 2. We show that when instruction-tuned in a 4K context length with only 500 steps, SAMBA can be extrapolated to a 256K context length with perfect memory recall in Passkey Retrieval (Mohtashami & Jaggi, 2023). In contrast, the fine-tuned SWA-based model simply cannot recall memories beyond 4K length. We further demonstrate that the instruction-tuned SAMBA 3.8B model can achieve significantly better performance than the SWA-based models on downstream long-context summarization tasks, while still keeping its impressive performance on the short-context benchmarks. In a more challenging multiple key-value retrieval task, Phonebook (Jelassi et al., 2024), we demonstrate that instruction fine-tuning enables SAMBA to bridge the retrieval performance gap with full-attention models, while exhibiting significantly better extrapolation ability when retrieving phone numbers beyond the training context length. Finally, we perform extensive analyzes and ablation studies across model sizes up to 1.7B parameters to validate the architectural design of SAMBA. We also offer potential explanations for the effectiveness of our simple hybrid approach through the lens of attention/selection entropy. To the best of our knowledge, Samba is the first hybrid model showing that linear complexity models can be substantially better than state-of-the-art Transformer models on short-context tasks at large scale, while still being able to extrapolate to extremely long sequences under the perplexity metric.

## 2 METHODOLOGY

We explore different hybridization strategies consisting of the layers of Mamba, Sliding Window Attention (SWA), and Multi-Layer Perceptron (Shazeer, 2020; Dauphin et al., 2016). We conceptualize the functionality of Mamba as the capture of recurrent sequence structures, SWA as the precise retrieval of memory, and MLP as the recall of factual knowledge. We also explore other linear recurrent layers including Multi-Scale Retention (Sun et al., 2023) and GLA (Yang et al., 2023) as potential substitutions for Mamba in Section 3.2. Our goal of hybridization is to harmonize between these distinct functioning blocks and find an efficient architecture for language modeling with unlimited length extrapolation ability.

### 2.1 ARCHITECTURE

As illustrated in Figure 1, we explore three kinds of layerwise hybridization strategies on the 1.7B scale: Samba, Mamba-SWA-MLP, and Mamba-MLP. We also explore other hybridization approaches with full self-attention on smaller scales in Section 4. The number of layers $N$ is set to 48 for Samba, Mamba-MLP, and Mamba, while Mamba-SWA-MLP has 54 layers, so each model has approximately 1.7B parameters. We only modify the layer-level arrangement for each of the models and keep every other configuration the same to have apple-to-apple comparisons. More details on the configuration of each layer are explained in the following subsections.

#### 2.1.1 MAMBA LAYER

Mamba (Gu & Dao, 2023) is a recently proposed SSM-based model with selective state spaces. It enables input-dependent gating to both the recurrent states and the input representation for a soft

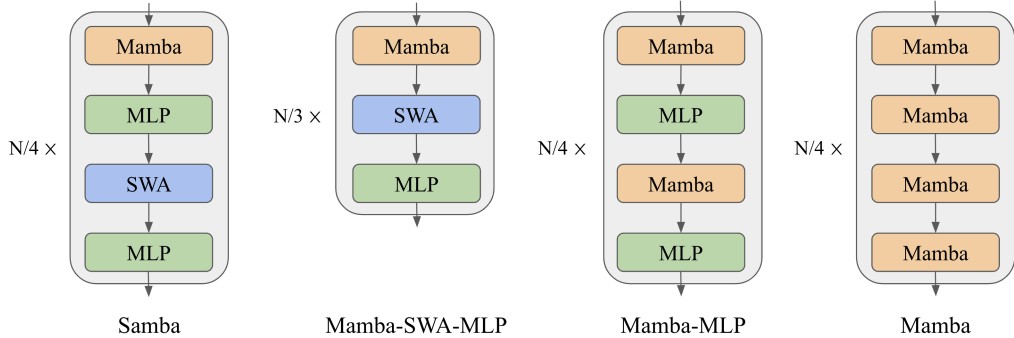

Figure 1: From left to right: Samba, Mamba-SWA-MLP, Mamba-MLP, and Mamba. The illustrations depict the layer-wise integration of Mamba with various configurations of Multi-Layer Perceptrons (MLPs) and Sliding Window Attention (SWA). We assume the total number of intermediate layers to be $N$, and omit the embedding layers and output projections for simplicity. Pre-Norm (Xiong et al., 2020; Zhang & Sennrich, 2019) and skip connections (He et al., 2016) are applied for each of the intermediate layers.

selection of the input sequence elements. Given an input sequence representation $\mathbf{X} \in \mathbb{R}^{n \times d_m}$, where $n$ is the length of the sequence and $d_m$ is the hidden size, Mamba first expands the inputs to a higher dimension $d_e$, *i.e.*,

$$\mathbf{H} = \mathbf{X}\mathbf{W}_{\text{in}} \in \mathbb{R}^{n \times d_e}$$

where $\mathbf{W}_{\text{in}} \in \mathbb{R}^{d_m \times d_e}$ is a learnable projection matrix. Then a Short Convolution (SC) (Poli et al., 2023) operator is applied to smooth the input signal,

$$\mathbf{U} = \text{SC}(\mathbf{H}) = \text{SiLU}(\text{DepthwiseConv}(\mathbf{H}, \mathbf{W}_{\text{conv}})) \in \mathbb{R}^{n \times d_e} \tag{1}$$

where $\mathbf{W}_{\text{conv}} \in \mathbb{R}^{k \times d_e}$ and the kernel size $k$ is set to 4 for hardware-aware efficiency. The Depthwise Convolution (He et al., 2019) is applied over the sequence dimension followed by a SiLU (Elfwing et al., 2017) activation function. The selective gate is then calculated through a low-rank projection followed by Softplus (Zheng et al., 2015),

$$\Delta = \text{Softplus}(\mathbf{U}\mathbf{W}_{\text{r}}\mathbf{W}_{\text{q}} + \mathbf{b}) \in \mathbb{R}^{n \times d_e} \tag{2}$$

where $\mathbf{W}_{\text{r}} \in \mathbb{R}^{d_e \times d_r}$, $\mathbf{W}_{\text{q}} \in \mathbb{R}^{d_r \times d_e}$ and $d_r$ is the low-rank dimension. $\mathbf{b} \in \mathbb{R}^{d_e}$ is carefully initialized so that $\Delta \in [\Delta_{\min}, \Delta_{\max}]$ after the initialization stage. We set $[\Delta_{\min}, \Delta_{\max}] = [0.001, 0.1]$, and find that these values are not sensitive to language modeling performance under the perplexity metric. The input dependence is also introduced for the parameters $\mathbf{B}$ and $\mathbf{C}$ of SSM,

$$\mathbf{B} = \mathbf{U}\mathbf{W}_{\text{b}} \in \mathbb{R}^{n \times d_s}$$

$$\mathbf{C} = \mathbf{U}\mathbf{W}_{\text{c}} \in \mathbb{R}^{n \times d_s}$$

where $d_s$ is the state dimension. For each time step $1 \leq t \leq n$, the recurrent inference of the Selective SSM (S6) is performed in an expanded state space $\mathbf{Z}_t \in \mathbb{R}^{d_e \times d_s}$, *i.e.*,

$$\mathbf{Z}_t = \exp(-\Delta_t \odot \exp(\mathbf{A})) \odot \mathbf{Z}_{t-1} + \Delta_t \odot (\mathbf{B}_t \otimes \mathbf{U}_t) \in \mathbb{R}^{d_e \times d_s}$$

$$\mathbf{Y}_t = \mathbf{Z}_t\mathbf{C}_t + \mathbf{D} \odot \mathbf{U}_t \in \mathbb{R}^{d_e}$$

where $\mathbf{Z}_0 = \mathbf{0}$, $\odot$ means the point-wise product, $\otimes$ means the outer product and $\exp$ means the point-wise natural exponential function. $\mathbf{D} \in \mathbb{R}^{d_e}$ is a learnable vector initialized as $D_i = 1$ and $\mathbf{A} \in \mathbb{R}^{d_e \times d_s}$ is a learnable matrix initialized as $A_{ij} = \log(j), 1 \leq j \leq d_s$, following the S4D-Real (Gu et al., 2022) initialization. In practice, Mamba implements a hardware-aware parallel scan algorithm for efficient parallelizable training. The final output is obtained through a gating mechanism similar to Gated Linear Unit (Shazeer, 2020; Dauphin et al., 2016),

$$\mathbf{O} = \mathbf{Y} \odot \text{SiLU}(\mathbf{X}\mathbf{W}_{\text{g}})\mathbf{W}_{\text{out}} \in \mathbb{R}^{n \times d_m}$$

where $\mathbf{W}_g \in \mathbb{R}^{d_m \times d_e}$ and $\mathbf{W}_{\text{out}} \in \mathbb{R}^{d_e \times d_m}$ are learnable parameters. In this work, we set $d_e = 2d_m$, $d_r = d_m/16$, and $d_s = 16$. The Mamba layer in SAMBA is expected to capture the time-dependent semantics of the input sequence through its recurrent structure. The input selection mechanism in the Mamba layer enables the model to focus on relevant inputs, thereby allowing the model to memorize important information in the long term.

### 2.1.2 SLIDING WINDOW ATTENTION (SWA) LAYER

We include Sliding Window Attention (Beltagy et al., 2020) layers to address the limitations of Mamba layers in capturing non-recurrent dependencies in sequences. Our SWA layer operates on a window size $w = 2048$ that slides over the input sequence, ensuring that the computational complexity remains linear with respect to the sequence length. RoPE (Su et al., 2021) is applied within the sliding window, with a base frequency of 10,000. By directly accessing the contents in the context window through attention, the SWA layer can retrieve high-definition signals from the middle to short-term history that cannot be clearly captured by the recurrent states of Mamba. We use FlashAttention 2 (Dao, 2023) for the efficient implementation of self-attention throughout this work. We also choose the 2048 sliding window size for efficiency consideration; FlashAttention 2 has the same training speed as Mamba's selective parallel scan at the sequence length of 2048 based on the measurements in (Gu & Dao, 2023).

### 2.1.3 MULTI-LAYER PERCEPTRON (MLP) LAYER

The MLP layers in SAMBA serve as the architecture's primary mechanism for nonlinear transformation and recall of factual knowledge (Dai et al., 2022). We use SwiGLU (Shazeer, 2020) for all the models trained in this paper and denote its intermediate hidden size as $d_p$. As shown in Figure 1, Samba applies separate MLPs for different types of information captured by Mamba and the SWA layers.

## 3 EXPERIMENTS AND RESULTS

We pre-train four SAMBA models with different parameter sizes, 421M, 1.3B, 1.7B and 3.8B, to investigate its performance across different scales. The details of the hyperparameters for the training and architecture designs are shown in Table 12 of Appendix G. We also train other hybrid architectures as mentioned in Section 2.1, including the baseline Mamba (Gu & Dao, 2023), Llama-3 (MetaAI, 2024; Dubey et al., 2024), and Mistral (Jiang et al., 2023) architecture on a scale of around 1.7B, with detailed hyperparameters in Table 11 of Appendix G. We do comprehensive downstream evaluations on a wide range of benchmarks, focusing on four main capabilities of the models: commonsense reasoning (ARC (Clark et al., 2018), PIQA (Bisk et al., 2020), WinoGrande (Sakaguchi et al., 2021), SIQA (Sap et al., 2019)), language understanding (HellaSwag (Zellers et al., 2019), BoolQ (Clark et al., 2019), OpenbookQA (Mihaylov et al., 2018), SQuAD (Rajpurkar et al., 2016), MMLU (Hendrycks et al., 2021), MMLU-Pro (Wang et al., 2024), GPQA(Rein et al., 2023)), truthfulness (TruthfulQA (Lin et al., 2022)) and math and coding (GSM8K (Cobbe et al., 2021), MBPP (Austin et al., 2021), HumanEval (Chen et al., 2021)).

Table 1: Downstream performance comparison between Samba-3.8B-IT and Phi-3-mini-4K on both long-context and short-context tasks. We report 5-shot accuracy (averaged by category) for MMLU, 8-shot CoT (Wei et al., 2022) for GSM8K, 0-shot pass@1 for HumanEval, ROUGE-L for both GovReport and SQuALITY. † Results from the Phi-3 technical report (Abdin et al., 2024).

| Model | MMLU | GSM8K | HumanEval | GovReport | SQuALITY |
|---|---|---|---|---|---|
| Phi-3-mini-4K-instruct † | 68.8 | 82.5 | 58.5 | 14.4 | **21.6** |
| Samba-3.8B-IT | **71.9** | **87.6** | **62.8** | **18.9** | 21.2 |

### 3.1 LANGUAGE MODELING ON TEXTBOOK QUALITY DATA

We first present results from our largest 3.8B SAMBA model, trained on the same data set used by Phi3 (Abdin et al., 2024) with 3.2T tokens. We follow the same multiphase pretraining strategy as Phi3-mini, and apply both the original Phi-3-mini post-training recipe and the Phi3-mini-June-2024 recipe to produce our instruction-tuned SAMBA 3.8B models, *i.e.*, Samba-3.8B-IT and Samba-3.8B (June) respectively. We report comprehensive benchmark results of the Samba 3.8B base model and Samba-3.8B (June) in Appendix B. As shown in Table 1, we evaluate the downstream performance of Samba-3.8B-IT on both long-context summarization tasks (GovReport (Huang et al., 2021), SQuALITY (Wang et al., 2022)) and major short-context benchmarks (MMLU, GSM8K, HumanEval). We can see that Samba has substantially better performance than Phi-3-mini-4k-instruct on both the short-context (MMLU, GSM8K, HumanEval) and long-context (GovReport) tasks, while

still having the 2048 window size of its SWA layer and maintaining the linear complexity for efficient processing of long documents. Details of data statistics and evaluation setup for long context tasks are included in Appendix F.

Table 2: Downstream evaluation of the architectures trained on 230B tokens of the Phi2 dataset. We report the unnormalized accuracy for multiple choice tasks. GSM8K is evaluated with 5-shot examples while other tasks are in zero-shot. Best results are in bold, second best underlined.

| Benchmark | Llama-3 1.6B | Mistral 1.6B | Mamba 1.8B | Mamba-SWA-MLP 1.6B | Mamba-MLP 1.9B | SAMBA 1.7B |
|---|---|---|---|---|---|---|
| ARC-Easy | 76.85 | 77.02 | 77.99 | 76.68 | _78.91_ | **79.25** |
| ARC-Challenge | 43.26 | 44.20 | 45.22 | 46.16 | _47.35_ | **48.21** |
| PIQA | 76.66 | 75.79 | _77.31_ | 76.50 | **78.84** | 77.10 |
| WinoGrande | 70.01 | 70.72 | _73.40_ | **73.72** | 72.38 | 72.93 |
| SIQA | 51.23 | 52.00 | 53.12 | **55.12** | _54.30_ | 53.68 |
| HellaSwag | 46.98 | 47.19 | _49.80_ | 49.71 | **50.14** | 49.74 |
| BoolQ | 68.20 | 70.70 | _74.83_ | 74.74 | 73.70 | **75.57** |
| OpenbookQA | 34.00 | 32.80 | _36.60_ | 33.80 | 35.40 | **37.20** |
| SQuAD | 74.88 | 72.82 | 67.66 | _76.73_ | 63.86 | **77.64** |
| MMLU | 43.84 | 43.54 | 45.28 | _47.39_ | 43.68 | **48.01** |
| TruthfulQA (MC1) | 25.70 | 25.09 | 26.81 | 26.20 | _26.44_ | **27.78** |
| TruthfulQA (MC2) | 40.35 | 38.80 | 40.66 | _40.80_ | 40.04 | **41.62** |
| GSM8K | 32.68 | 32.45 | 32.07 | **44.05** | 27.52 | _38.97_ |
| MBPP | 46.30 | 47.08 | _47.86_ | 47.08 | 47.08 | **48.25** |
| HumanEval | 36.59 | 36.59 | 35.98 | _37.80_ | 31.10 | **39.02** |
| **Average** | 51.17 | 51.12 | 52.31 | _53.77_ | 51.38 | **54.33** |

To examine the different hybridization strategies mentioned in Section 2.1, we train 6 models with around 1.7B parameters on the Phi2 (Li et al., 2023) dataset with 230B tokens and evaluate them in the full suite of 15 downstream benchmarks to have a holistic assessment of hybrid and purebred architectures. As shown in Table 2, SAMBA demonstrates superior performance on a diverse set of tasks, including commonsense reasoning (ARC-Challenge), language understanding (MMLU, SQuAD), TruthfulQA and code generation (HumanEval, MBPP). It outperforms both the pure attention-based and SSM-based models in most tasks and achieves the best average performance. By comparing the performance of Mamba-MLP and Mamba in Table 2, we can observe that replacing Mamba blocks with MLPs does not harm common sense reasoning ability, but its performance in language understanding and complex reasoning ability, such as coding and mathematical reasoning, degenerates significantly. We can also see that pure Mamba models fall short on retrieval intensive tasks such as SQuAD due to their lack of precise memory retrieval ability. The best results are achieved through the combination of the attention and Mamba modules, as shown with our Samba architecture. We can also notice that Mamba-SWA-MLP has significantly better performance on GSM8K, potentially resulting from a closer collaboration between the Mamba and the SWA layers. The distinct downstream performances of different hybridization strategies pose interesting future work for developing task-adaptive dynamic architectures.

## 3.2 EXPLORATION ON HYBRIDIZING ATTENTION AND LINEAR RECURRENCE

Since SSMs belong to a broader realm of linear recurrent models (Orvieto et al., 2023; Qin et al., 2023; Yang et al., 2023; Katsch, 2023; Qin et al., 2024; Yang et al., 2024), there exist multiple alternatives other than Mamba when combing attention-based layers with recurrent neural networks. We also add architecture ablation studies to justify the design choices of Samba. Specifically, in addition to Llama-2, Mamba, Samba and Mamba-SWA-MLP, we investigate the comparative analysis of the following architectures:

- **Llama-2-SWA** is a pure attention-based architecture that replaces all full attention layers in Llama-2 with sliding window attention.

Table 3: Perplexity on the validation set of SlimPajama for different attention and linear recurrent model architectures trained at 4,096 context length. We use window size 2,048 for Sliding Window Attention (SWA). The perplexity results have a fluctuation around $\pm 0.3\%$.

| Architecture | Size | Layers | Training Speed ($\times 10^5$ tokens/s) | Validation Context Length | | |
|---|---|---|---|---|---|---|
| | | | | 4096 | 8192 | 16384 |
| *20B training tokens on 8×A100 GPUs* | | | | | | |
| Llama-2 | 438M | 24 | 4.85 | 11.14 | 47.23 | 249.03 |
| Llama-2-SWA | 438M | 24 | 4.96 | 11.12 | 10.66 | 10.57 |
| Mamba | 432M | 60 | 2.46 | 10.70 | 10.30 | 10.24 |
| Sliding GLA | 438M | 24 | 4.94 | 10.43 | 10.00 | 9.92 |
| Sliding RetNet | 446M | 24 | 4.32 | 10.38 | 9.96 | 9.87 |
| Mega-S6 | 422M | 24 | 3.26 | 12.63 | 12.25 | 12.25 |
| Mamba-SWA-MLP | 400M | 24 | 4.21 | 10.07 | 9.67 | 9.59 |
| MLP2-SWA-MLP | 417M | 24 | **5.08** | 10.95 | 10.50 | 10.41 |
| SAMBA-NoPE | 421M | 24 | 4.48 | 10.11 | 28.97 | 314.78 |
| SAMBA | 421M | 24 | 4.46 | **10.06** | **9.65** | **9.57** |
| *100B training tokens on 64×H100 GPUs* | | | | | | |
| Llama-2 | 1.3B | 40 | 25.9 | 7.60 | 44.32 | 249.64 |
| Llama-2-SWA | 1.3B | 40 | 26.2 | 7.60 | 7.37 | 7.21 |
| Mamba | 1.3B | 48 | 17.8 | 7.47 | 7.26 | 7.15 |
| Sliding GLA | 1.2B | 36 | 25.9 | 7.58 | 7.35 | 7.19 |
| Sliding RetNet | 1.4B | 36 | 23.0 | 7.56 | 7.35 | 7.56 |
| Mega-S6 | 1.3B | 36 | 17.9 | 9.01 | 8.81 | 8.68 |
| Mamba-SWA-MLP | 1.3B | 36 | 23.5 | 7.37 | 7.16 | 7.00 |
| MLP2-SWA-MLP | 1.3B | 36 | **26.6** | 7.81 | 7.58 | 7.42 |
| SAMBA-NoPE | 1.3B | 36 | 25.2 | 7.33 | 20.40 | 326.17 |
| SAMBA | 1.3B | 36 | 25.2 | **7.32** | **7.11** | **6.96** |

- **Sliding RetNet** replaces Mamba layers in the Samba architecture with Multi-Scale Retention (Sun et al., 2023) layers. RetNet is a linear attention model with fixed and input-independent decay applying to the recurrent hidden states.

- **Sliding GLA** replaces Mamba layers in the Samba architecture with Gated Linear Attention (GLA) (Yang et al., 2023). GLA is a more expressive variant of linear attention with input-dependent gating.

- **Mega-S6** replaces all MD-EMA modules in the Mega (Ma et al., 2023) architecture with the ShortConv+S6 combinations from Mamba to adapt Mega to the modern Mamba architecture. Rotary position embedding, RMSNorm and Softmax attention are also adopted. We set the intermediate dimension of the Mega-S6 layer to be $d_m$ so that it has a roughly $5d_m^2$ number of parameters. This represents a classical baseline that conducts sequential intra-layer SSM-Attention hybridization.

- **MLP2-SWA-MLP** replaces all Mamba layers in the Samba architecture to SwiGLU layers with $6d_m^2$ number of parameters.

- **Samba-NoPE** removes the rotary relative position embedding in Samba and does not have any position embedding in the architecture.

We pre-train all models on the same SlimPajama (Soboleva et al., 2023) dataset under both around 438M and 1.3B settings, and evaluate these models by calculating perplexity on the validation set with context length at 4096, 8192, and 16384 tokens to investigate their zero-shot length extrapolation ability. Peak training throughput is also measured as an efficiency metric. The details of the hyperparameter settings are included in Appendix G. As shown in Table 3, SAMBA consistently outperforms all other models in different context lengths and model sizes. The training speed of SAMBA is competitive compared to pure Transformer-based models on the 1.3B scale. Mamba has significantly worse training throughput because Mamba layers have slower training speed than MLP layers, and the purebred Mamba models need to have more layers than other models at the same number of parameters. Comparing Mamba-SWA-MLP with Samba, we can see that Samba has slightly better perplexity scores and higher training throughput. Mamba-SWA-MLP trades off the MLP layers with more I/O intensive Mamba and Attention layers, leading to slower training speed.

This also indicates that Mamba-SWA-MLP will have slower decoding speed than Samba due to larger total cache size resulting from more SSMs and Attention layers. We can further observe that replacing Mamba with MLP speeds up the training but harms perplexity significantly, indicating the importance of Mamba layers in the Samba architecture. Interestingly, even though we use SWA in Samba architecture, Samba-NoPE still has exploded perplexities beyond its training length without RoPE. We can also find that while RetNet can extrapolate well under the 438M scale, it has an increasing perplexity on 16K length at the 1.4B scale, which may indicate that its input-independent decay may need specific tuning at different scales to work well.

Table 4: Downstream evaluation of models pre-trained with 100B tokens from SlimPajama. We measure the character-normalized accuracy for HellaSwag following Gu & Dao (2023). All tasks are evaluated in zero-shot.

| Architecture | Size | ARC-Easy acc ↑ | HellaSwag acc_norm ↑ | Wino. acc ↑ | PIQA acc ↑ | LAMBADA acc ↑ | Avg. |
|---|---|---|---|---|---|---|---|
| LLaMA-2 | 1.3B | 55.09 | 52.32 | 53.35 | 71.11 | 48.52 | 56.08 |
| LLaMA-2-SWA | 1.3B | 56.65 | 52.59 | 54.93 | 71.60 | 47.56 | 56.67 |
| Sliding GLA | 1.2B | 56.94 | 52.52 | **56.75** | 71.38 | 48.17 | 57.15 |
| Sliding RetNet | 1.4B | 57.66 | 52.64 | **56.75** | 71.33 | 48.34 | 57.34 |
| Mega-S6 | 1.3B | 50.63 | 41.91 | 52.96 | 68.17 | 37.88 | 50.31 |
| Mamba | 1.3B | 58.08 | **54.93** | 53.99 | 71.98 | 45.97 | 56.99 |
| Mamba-SWA-MLP | 1.3B | **59.64** | 54.50 | 55.25 | **72.42** | 49.12 | 58.19 |
| MLP2-SWA-MLP | 1.3B | 55.18 | 50.32 | 52.80 | 70.67 | 48.11 | 55.42 |
| SAMBA-NoPE | 1.3B | 58.38 | 54.62 | 56.51 | 72.03 | 51.08 | 58.52 |
| SAMBA | 1.3B | 58.21 | 54.73 | 55.72 | 72.36 | **51.68** | **58.54** |

In Table 4, we evaluate all our 1.3B scale models on five typical commonsense reasoning tasks (ARC-Easy, HellaSwag, WinoGrande, PIQA and the OpenAI variant[1] of LAMBADA (Paperno et al., 2016) ) to understand the effect of architecture designs on downstream performances. We can see that Samba has the best average accuracy, outperforming the LLaMA 2 architectures by a large margin. Similar to our perplexity evaluation, Samba and Samba-NoPE have similar average accuracies, whereas Mamba-SWA-MLP falls slightly behind. We observe that different architectures excel at different tasks. Mamba-SWA-MLP performs best on ARC-Easy, while Samba and Samba-NoPE achieve superior results on LAMBADA. Hybrid models based on Mamba generally outperform hybrid linear attention models and pure softmax-attention models on HellaSwag.

## 3.3 EFFICIENT LENGTH EXTRAPOLATION

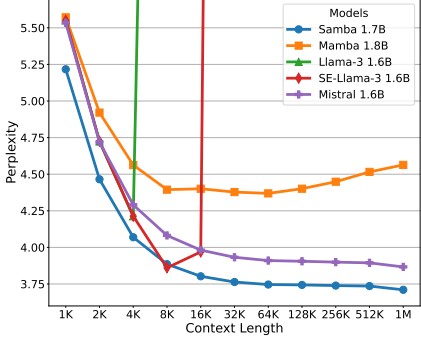
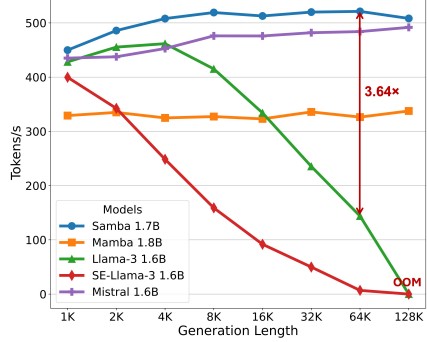

(a) Perplexity on the test set of Proof-Pile     (b) Decoding throughput with batch size 16

Figure 2: SAMBA shows improved prediction up to 1M tokens in the Proof-Pile test set while achieving a 3.64× faster decoding throughput than the Llama-3 architecture on 64K generation length. We also include an SE-Llama-3 1.6B baseline which applies the SelfExtend (Jin et al., 2024) approach for zero-shot length extrapolation. All models are trained with 4K sequence length.

We use the test split of the Proof-Pile (Zhangir Azerbayev & Piotrowski, 2022) dataset to evaluate the length extrapolation ability of our models at a scale of around 1.7B parameters. We follow Position

---

[1] https://huggingface.co/datasets/EleutherAI/lambada_openai

Interpolation (Chen et al., 2023a) for data pre-processing. The sliding window approach (Press et al., 2021) is used for the perplexity evaluation with a window size of 4096. Besides having the decoding throughput in Figure 2 for the generation efficiency metric, we also measure the prompt processing speed in Figure 6 of Appendix B for the models SAMBA 1.7B, Mistral 1.6B, Mamba 1.8B, Llama-3 1.6B and its Self-Extended (Jin et al., 2024) version SE-Llama-3 1.6B with the prompt length sweeping from 1K to 128K. We set the group size to 4 and the neighborhood window to 1024 for Self-Extension. We fix the total processing tokens per measurement to be 128K and varying the batch size accordingly. The throughput is measured on a single A100 GPU with the precision of bfloat16. We repeat the measurements 10 times and report the averaged results. We can see that Samba achieves $3.73\times$ higher throughput in prompt processing compared to Llama-3 1.6B at the 128K prompt length, and the processing time remains linear with respect to the sequence length. We can also observe that the existing zero-shot length extrapolation technique introduces significant inference latency overhead on the full-attention counterpart, while it still cannot extrapolate infinitely with perplexity performance comparable to that of Samba. In Figure 2, we can also see that Mamba has a slowly and stably increasing perplexity up to 1M sequence length, which indicates that linear recurrent models can still not extrapolate infinitely if the context length is extremely large.

## 3.4 LONG-CONTEXT UNDERSTANDING

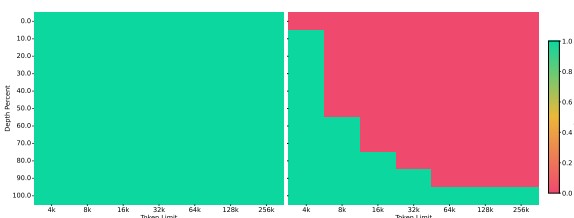

Figure 3: Passkey Retrieval performance up to 256K context length for SAMBA 1.7B (Left) vs. Mistral 1.6B (right) instruction tuned on 4K sequence length with 500 steps.

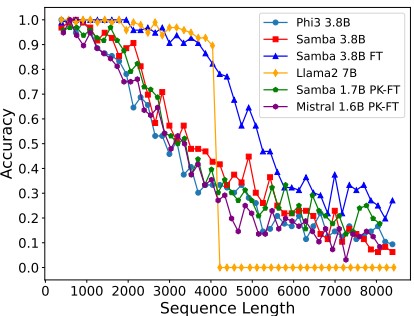

Figure 4: Phonebook evaluation accuracy of different base models.

Beyond its efficiency in processing long context, Samba can also extrapolate its memory recall ability to 256K context length through supervised fine-tuning, and still keeps its linear computation complexity. We fine-tune Samba 1.7B on Passkey Retrieval with a 4K training sequence length for only 500 steps. As presented in Figure 3, SAMBA 1.7B demonstrates a remarkable ability to recall information from significantly longer contexts compared to Mistral 1.6B, a model based solely on Sliding Window Attention (SWA). This capability is particularly evident in the heatmap, where SAMBA maintains the perfect retrieval performance across a wider range of pass-key positions in a long document of up to 256K length. We also draw the training loss curve and the overall passkey retrieval accuracy across the fine-tuning procedure in Figure 7 and Figure 8 of Appendix C. We find that despite the fact that both architectures can reach near-zero training loss in less than 250 steps, Samba can achieve near-perfect retrieval early at 150 training steps, while the Mistral architecture struggles at around 30% accuracy throughout the training process. This shows that Samba can have better long-range retrieval ability than SWA due to the input selection mechanism introduced by the Mamba layers. In Figure 8, we can also notice that the pre-trained base Samba model has a retrieval accuracy (at step 0) similar to that of Mistral, highlighting the need for future work to improve Samba's zero-shot retrieval capabilities.

The encouraging results on Passkey Retrieval drives us to further explore the limits of our finetuning approach. We perform instruction tuning to the Samba-3.8B base model on Phonebook (Jelassi et al., 2024) with only 100 steps on 4K sequence length and evaluate the resulting Samba-3.8B-FT model for a sequence length up to 8K. The evaluation setting requires the models to retrieve a random phone number from a phone book containing 20 (length 400) to 480 (length 8400) name-number pairs, resulting in a pressure test of memorization to Samba which has a constant memory state size. Surprisingly, as shown in Figure 4, we can see that the Samba-3.8B-FT model can close most of its gap with a full-attention model (Llama2 7B) that has twice the parameter size within the 4K training length, and achieves much better extrapolation accuracy compared to all other models including

the Phi3 base model which also uses 2K sliding window attention. Since both Passkey Retrieval and Phonebook require models to remember numbers in a long context document, it is interesting to investigate if a model instruction-tuned on one task can transfer its ability to the other task in zero-shot. We directly evaluate the Passkey Retrieval finetuned Samba 1.7B and Mistral 1.6B models (named Samba 1.7B PK-FT and Mistral 1.6B PK-FT respectively) on the Phonebook task. As shown in Figure 4, Samba 1.7B has slightly better retrieval accuracy than Mistral 1.6B, but both models cannot generalize their number recall ability beyond its sliding window size. We leave it for future work to further explore the transferability of long-context capabilities in linear complexity models.

## 4 ANALYSIS

In this section, we analyze the experimental results of SAMBA by answering the following research questions. The perplexity results on SlimPajama have a fluctuation around $\pm 0.3\%$. Training speed is measured on 8×H100 GPUs by default. All the models in this section are trained on SlimPajama with 20B tokens and 4K sequence length, unless otherwise specified. We also have additional analyses on the training of SWA-based models and the effectiveness of short convolution in Appendix D.

**Why not hybridize with full attention?**  Some previous works (Fu et al., 2023; Lieber et al., 2024) suggest a hybrid architecture of Mamba with full attention. However, as shown in Table 5, the extrapolation perplexity is exploding at a context length of 16K even if a single full attention layer is placed at the beginning of the model. Although hybridization with full attention in the second and middle sixth blocks (the fourth row in the table), following Dao et al. (2022b), can bridge the perplexity gap between full-attention hybrids and Samba, they still cannot extrapolate beyond the training sequence lengths. Samba also has much better training throughput compared to Mamba-MLP alternatives because self-attention with the FlashAttention 2 implementation is more training efficient than Mamba when the sequence length is 4096.

Table 5: Perplexity on SlimPajama of Mamba-MLP architectures with full attention layers replacing Mamba layers at different block indices. We define a block as two consecutive layers with a Mamba/Attention layer followed by an MLP. All the models have 12 blocks in total.

| Architecture | Size | Block Index of Full Attention | Training Speed ($\times 10^5$ tokens/s) | Validation Context Length | | |
|---|---|---|---|---|---|---|
| | | | | 4096 | 8192 | 16384 |
| Mamba-MLP | 449M | 11 | 7.78 | 10.29 | 10.53 | 13.66 |
| | 449M | 5 | 7.78 | 10.10 | 10.05 | 12.83 |
| | 449M | 0 | 7.78 | 10.89 | 10.55 | 10.63 |
| | 443M | 1, 5 | 7.93 | **10.06** | 10.34 | 13.57 |
| SAMBA | 421M | SWA at odd indices | 8.59 | **10.06** | **9.65** | **9.57** |

**How many parameters should be allocated to Attention?**  Given that Mamba can already capture low-rank information in the sequences through recurrent compression, the attention layers in Samba theoretically will only need to focus on information retrieval where a small number of attention heads should suffice. In Table 6, we explore the techniques of query head grouping (Ainslie et al., 2023; Shazeer, 2019), for both the Llama and Samba models. Surprisingly, both the Llama-2-SWA architecture and the Samba architecture show improved validation perplexity when there is only one key-value head. We conjecture that this is because small language models can be more easily optimized with fewer KV heads to pay attention to the contexts. We can also see that Samba has a $2\times$ smaller optimal number of query heads than the SWA model, which confirms our hypothesis that Samba can support a smaller number of attention heads.

**Potential explanations on why hybrid is better?**  We examine the entropy of the attention distributions for both the Samba 1.7B and the Mistral 1.6B models. As shown in Figure 5a, the Samba model has a larger variance of the attention entropy distributed over the layer indices, with an interesting pattern that the upper and lower layers have entropy higher than the middle layers. This may indicate that the attention layers are more specialized in the Samba architecture, with the middle layers focusing on precise retrieval with low-entropy attention, and the top and bottom layers focusing on integrating the global information through high-entropy attention. We can also see in Figure 5b that,

Table 6: Perplexity on SlimPajama of Llama-2-SWA and Samba models at the 430M scales trained with different number of Query and Key-Value heads. "KV Size" means the size of Key-Value vectors per token and attention layer. Since grouped query attention will reduce the parameters for attention from $4d_m^2$ to roughly $2d_m^2$, we increase the intermediate size of MLP from $8/3d_m$ to $3d_m = 4608$ to have roughly the same number of total parameters as the original models.

| Query Head | Key-Value Head | Head Dim. | KV Size | Model Size | Training Speed ($\times 10^5$ tokens/s) | Validation Context Length | | |
|---|---|---|---|---|---|---|---|---|
| | | | | | | 4096 | 8192 | 16384 |
| *Llama-2-SWA Architecture* | | | | | | | | |
| 12 | 2 | 128 | 512 | 419M | 10.01 | 11.11 | 10.64 | 10.56 |
| 6 | 1 | 256 | 512 | 419M | 9.98 | 11.09 | 10.62 | 10.54 |
| 12 | 1 | 128 | 256 | 414M | 10.25 | **10.89** | **10.44** | **10.35** |
| 12 | 4 | 128 | 1024 | 428M | 9.85 | 11.11 | 10.64 | 10.56 |
| *Samba Architecture* | | | | | | | | |
| 12 | 2 | 128 | 512 | 426M | 8.55 | 10.09 | 9.68 | 9.60 |
| 6 | 1 | 256 | 512 | 426M | 8.46 | **9.99** | **9.59** | **9.51** |
| 12 | 1 | 128 | 256 | 424M | 8.62 | 10.07 | 9.66 | 9.58 |
| 12 | 4 | 128 | 1024 | 431M | 8.57 | 10.02 | 9.62 | 9.55 |

compared to the Mamba-MLP model, Samba has a higher entropy of input selection probabilities in the middle layers. This indicates that, given the memory recalling ability of the attention layers, the Mamba layers can focus more on modeling the recurrent structure rather than performing retrieval with precise input selections. This kind of specialization can be beneficial for the downstream model performance, which may explain the impressive results from the Samba architecture. Details on how entropy is calculated are included in Appendix E.

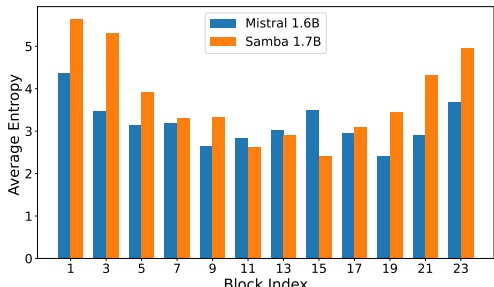
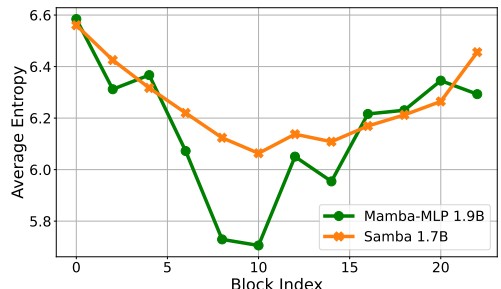

(a) Average attention entropy per decoding step    (b) Average S6 selection entropy on full sequences

Figure 5: The average entropy of the attention mechanism and the Mamba's S6 input selection mechanism at each block of layers on 100 random samples from the GSM8K dataset.

## 5 CONCLUSION

In this paper, we introduce SAMBA, a simple yet powerful hybrid neural architecture designed for efficient language modeling with unlimited context length. We show that SAMBA substantially outperforms state-of-the-art pure attention-based and SSM-based models across a wide range of benchmarks including common-sense reasoning, language understanding, mathematics and coding. Furthermore, SAMBA exhibits remarkable efficiency in processing long contexts, achieving substantial speedups in prompt processing and decoding throughput compared to the state-of-the-art Transformer architecture. The architecture's ability to extrapolate memory recall to very long contexts (up to 256K) through minimal fine-tuning underscores its practical applicability for real-world tasks requiring extensive context understanding. This efficient long-term memorization ability is further demonstrated to be useful by our evaluations in downstream long-context summarization tasks. Our analyses also provide insight into the optimal training configurations for hybrid models and underscore the benefits of combining attention mechanisms with SSMs. We find that allocating fewer parameters to the attention mechanism while leveraging Mamba's strengths for capturing recurrent structures leads to more efficient and effective language modeling. Our results suggest that SAMBA is a strong neural architecture for language modeling with unlimited context length.

## ACKNOWLEDGEMENT

We want to thank Shuohang Wang and Liyuan Liu for helping with the training infrastructure, Mojan Javaheripi and the team for the pre-training data, Ziyi Yang, Jianwen Zhang, Junheng Hao and the team for helping with post-training. The first author also wants to thank Songlin Yang for her Triton implementation of Mamba.

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

# A    RELATED WORKS

**Hybrid Recurrent Models**    Many recent works (Park et al., 2024; Jelassi et al., 2024; Akyürek et al., 2024) point out the lack of retrieval ability of linear SSMs, and propose hybridization of SSMs with the Attention mechanism. However, the history of SSM/RNN-Attention hybridization can be directly dated back to the birth of the Attention mechanism (Bahdanau et al., 2014) which is proposed as a soft feature alignment technique for recurrent models to cope better with long sequences. The revitalization of the fact that linear recurrent models are sequentially parallelizable (Martin & Cundy, 2018; Gu et al., 2021) has catalyzed a contemporary renaissance in hybrid recurrent architectures. SPADE (Zuo et al., 2022), GSS (Mehta et al., 2023), MEGA (Ma et al., 2023), Block State transformers (Fathi et al., 2023) and Megalodon (Ma et al., 2024) combine SSMs with chunked attention, while H3 (Dao et al., 2022b), Mambaformer (Park et al., 2024) and Jamba (Lieber et al., 2024; Team et al., 2024) propose to hybridize with quadratic self-attention. Our works focus particularly on the wall-time efficiency and the length extrapolatability of the hybrid SSM-Attention models, and propose to interleave SSMs with Sliding Window Attention (SWA), which has both linear computation complexity and the translation-invariant property over the sequence length. Infini-Attention (Munkhdalai et al., 2024) is a recently proposed method that implements an intra-layer hybridization (Wu et al., 2022) between SWA and Linear Attention with the delta rule (Schlag et al., 2021). While the preliminary results look promising, its performance in the setting of large-scale pre-training from scratch remains questionable. The most similar work to ours is Griffin (De et al., 2024), which interleaves the Real-Gated Linear Recurrent Unit (RG-LRU) with Sliding Window Attention (SWA). However, Samba hybridizes SWA with Mamba instead of RG-LRU and shows that this simple hybrid architecture can provide substantially better performance over state-of-the-art Transformer architectures across scales, while Griffin and its follow-up work RecurrentGemma (Botev et al., 2024) only show comparable or worse results than Transformers. The original Mamba paper (Gu & Dao, 2023) also explores hybridizing pure Mamba models with full attention or MLP layers, but it does not consider the wall-time efficiency of these hybridization and only achieves marginally better performance than the pure Mamba model. In contrast, we are the first to show that interleaving Mamba with both SWA and MLP can substantially outperform modern Transformers (and Mamba) at a scale up to 3.8B parameters, while achieving comparable training speed and better length extrapolation ability under the perplexity metrics.

**Efficient Sparse Attention**    Previous works have proposed sparsifying self-attention (Vaswani et al., 2017) with a static attention pattern (Child et al., 2019; Zaheer et al., 2020; Beltagy et al., 2020) or a dynamic learnable pattern (Roy et al., 2020; Kitaev et al., 2020; Ren et al., 2023) to model long sequences with subquadratic complexity over the sequence length. However, due to the lack of hardware-aware efficient implementation, its actual wall-time training efficiency is often worse than the dense attention optimized with FlashAttention (Dao et al., 2022a; Dao, 2023; Shah et al., 2024). In this work, we choose Sliding Window Attention, a simple static sparse attention pattern, because it can easily leverage the highly optimized FlashAttention kernels to enjoy an actual training speed-up over its dense self-attention counterpart.

**Length Extrapolation**    Many previous works have focused on extending the context length of pretrained Transformers to improve their performance on long-context tasks. Methods such as LM-Infinite (Han et al., 2023), StreamingLLM (Xiao et al., 2024), and LongLoRA (Chen et al., 2023b) achieve linear complexity for length extrapolation, but they can only stabilize perplexity beyond the training sequence length rather than significantly improve it. In contrast, we demonstrate that pre-training Transformers with Sliding Window Attention from scratch enables natural improvements in perplexity beyond the training sequence length. Other approaches, including LLaMA-2-Long (Xiong et al., 2023), LongLLaMA (Tworkowski et al., 2023), PI (Chen et al., 2023a), LongRoPE (Ding et al., 2024) and Self-Extend (Jin et al., 2024), attempt to extend the full attention through modifying position embedding or continual training strategies, but they typically retain quadratic complexity in the attention mechanism with additional computation or memory I/O overhead, therefore they do not scale well to very long sequences. Although these methods achieve an improved perplexity on a sequence length that is multiple times longer than the training sequence length, their perplexity still explodes if the sequence is extremely long. Our method achieves both linear complexity and superior extrapolation performance compared to zero-shot length extrapolation methods, such as Self-Extend, under the perplexity metric. However, we acknowledge that, in terms of zero-shot

retrieval performance, our method still lags behind these approaches. This underscores a trade-off between perplexity and retrieval performance in length extrapolation, which we plan to explore and address in future work.

# B ADDITIONAL EVALUATION RESULTS

In Table 7, we conduct comprehensive evaluations on a diverse subset of benchmarks to assess SAMBA 3.8B base model's performance across all the domains mentioned in Section 3 to ensure a thorough examination of the model's capabilities. We also report the performance of the Transformer++ (TFM++) model, which uses the same architecture, pre-training recipe as Phi3-mini, for a fair comparison. The details of the generation configurations are included in Appendix G. We compare with several strong baselines, including Llama 2 (Touvron et al., 2023), Mistral (Jiang et al., 2023), Mamba (Gu & Dao, 2023), Gemma (Team, 2024), Recurrent-Gemma (R-Gemma) (Botev et al., 2024), Llama 3 (MetaAI, 2024) and TFM++. As shown in Table 7, SAMBA achieves the highest average score on all benchmarks, demonstrating its superior performance in handling various language comprehension tasks. Notably, SAMBA excels in the GSM8K benchmark, achieving an absolute 18.1% higher accuracy than TFM++ trained on the same dataset. This shows the surprising complementary effect of combining SSM with the attention mechanism. We conjecture that when combined with attention, Mamba, as an input-dependent SSM, can focus more on performing the arithmetic operation through its recurrent states than on doing the retrieval operation which can be easily learned by the sliding window attention.

Table 7: Downstream performance comparison of the SAMBA 3.8B base model with other pretrained base language models without instruction tuning. ARC-C and HellaSwag are measured with character-normalized accuracy. MMLU and GSM8K are measured in 5-shot, while others are in zero-shot. We report the MC2 score for TruthfulQA, maj@1 for GSM8K, and pass@1 for HumanEval. * Measured by ours. The fair comparison should only be considered between TFM++ and Samba.

| Model | Size | Tokens | MMLU | Hella-Swag | ARC-C | Wino-Gran. | Truth. QA | GSM 8K | Hum. Eval | Avg. |
|---|---|---|---|---|---|---|---|---|---|---|
| Llama 2 | 6.7B | 2T | 45.3 | 77.2 | 45.9 | 69.2 | 38.8 | 14.6 | 12.8 | 43.4 |
| | 13B | 2T | 54.8 | 80.7 | 49.4 | 72.8 | 37.4 | 28.7 | 18.3 | 48.9 |
| Mistral | 7.2B | - | 60.1 | **81.3** | 55.5 | 75.3 | 42.2 | 35.4 | 30.5 | 53.6 |
| Mamba | 2.8B | 600B | 26.2 | 71.0 | 41.7 | 65.9 | 34.4* | 3.6* | 7.3* | 35.7 |
| Gemma | 2.5B | 3T | 42.3 | 71.4 | 42.1 | 65.4 | 33.1 | 17.7 | 22.0 | 42.0 |
| | 8.5B | 6T | 64.3 | 81.2 | 53.2 | 72.3 | 44.8 | 46.4 | 32.3 | 56.4 |
| R-Gemma | 2.7B | 2T | 38.4 | 71.0 | 42.3 | 67.8 | 35.1 | 13.4 | 21.3 | 41.3 |
| Llama 3 | 8.0B | 15T+ | 66.6 | 79.2* | 53.2* | 72.6* | 43.9 | 45.8 | 28.7* | 55.8 |
| TFM++ | 3.8B | 3.2T | 67.2 | 76.6 | 53.8 | 72.6 | **47.3** | 51.5 | 51.8 | 60.1 |
| SAMBA | 3.8B | 3.2T | **71.2** | 77.4 | **55.7** | **77.1** | 43.4 | **69.6** | **54.9** | **64.2** |

As shown in Table 8, we can see that post-trained hybrid models can achieve superior performance compared to industry-standard Transformer-based LLMs such as Llama-3.1-Instruct 8B and Llama-3.2-Instruct 3B, and SSM-based LLMs such as FalconMamba[2]. Recent progress on hybrid LLMs, including Jamba 1.5 (Team et al., 2024) and our own work on SAMBA, shows significant improvement over earlier approaches like R-Gemma (Botev et al., 2024), which hybridizes attention with linear recurrent models but is trained on smaller data scales. SAMBA delivers comparable performance to Jamba-1.5-Mini while using around 3× fewer active parameters and 13× fewer total parameters, due to an advanced text-book data synthesis technique (Abdin et al., 2024). Additionally, SAMBA outperforms the Phi3 architecture, which is trained on the same data and optimization setting, further highlighting the superiority of our hybrid architecture over modern Transformer models.

Table 8: Post-trained models quality on representative benchmarks under the chat mode. The fair comparison should only be considered between SAMBA and Phi3 as we control the training recipes and datasets to be the same. Best results are in bold, second best underlined.

| Category | Benchmark | SAMBA (June) 3.8B | Phi3 (June) 3.8B | R-Gemma 9B | FalconMamba 7B | Jamba-1.5-Mini 12B/52B | Llama-3.2-In 3B | Llama-3.1-In 8B |
|---|---|---|---|---|---|---|---|---|
| MMLU | MMLU (5-shot) | 69.0 | 67.2 | 60.5 | 62.1 | **69.7** | 61.8 | 68.1 |
| | MMLU-Pro (0-shot, CoT) | **47.9** | 46.5 | 17.8 | 14.5 | 42.5 | 39.2 | 44 |
| Reasoning | ARC-C (10-shot) | **87.8** | 86.8 | 52.0 | 62.0 | 85.7 | 76.1 | 83.1 |
| | GPQA (0-shot, CoT) | 29.5 | 29.0 | 4.7 | 8.1 | **32.3** | 26.6 | 26.3 |
| Math | GSM8K (8-shot, CoT) | **86.4** | 84.8 | 42.6 | 52.5 | 75.8 | 75.6 | 77.4 |
| Code | HumanEval (0-shot) | **70.1** | 66.5 | 31.1 | - | 62.8 | 62.8 | 66.5 |
| | MBPP (3-shot) | 71.7 | 70.0 | 42.0 | - | **75.8** | 67.2 | 69.4 |
| Average | | **66.1** | 64.4 | 35.8 | - | 63.5 | 58.5 | 62.1 |

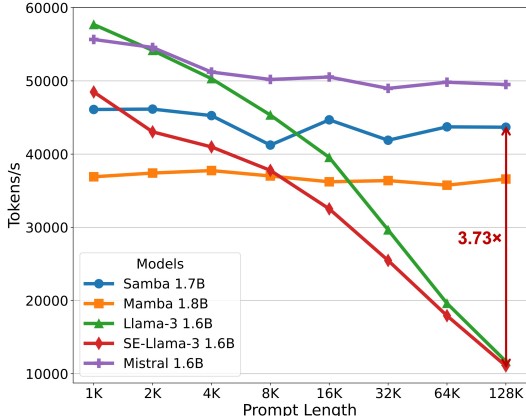

Figure 6: Prompt processing throughput of different models with around 1.7B parameters.

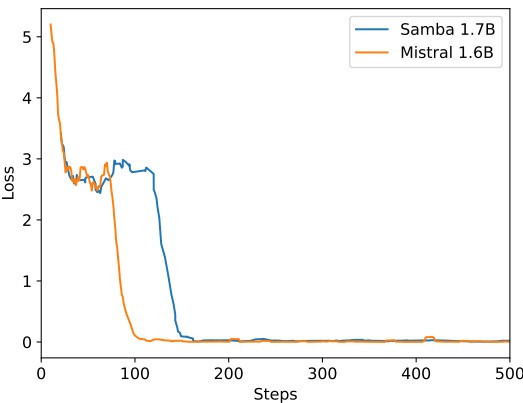

Figure 7: Training loss curves of Samba 1.7B and Mistral 1.6B models during 500 steps of instruction tuning on Passkey Retrieval with 4K sequence length. We plot the loss curves for both models using the simple moving average of window size 10.

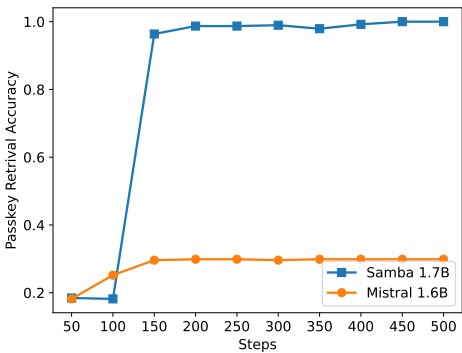

Figure 8: Overall passkey retrieval accuracy on the 256K document length of Samba 1.7B and Mistral 1.6B models during 500 steps of instruction tuning.

## C    ADDITIONAL EXPERIMENT DETAILS

We perform instruction tuning for both Mistral 1.6B and Samba 1.7B on Passkey Retrieval using document length 4096, where we generated the data on the fly through randomly sampling a 5-digit integer passkey value and a location/depth between zero and the document length to insert the passkey. The model is then asked to generate the passkey given the full document. We train both models using batch size 2048, 250 warm-up steps with a peak learning rate of $1e^{-4}$, and 0.1 weight decay with AdamW (Loshchilov & Hutter, 2018) optimizer. In both cases, the loss converges quickly in 100-200 steps. During the evaluation, we measure the overall average accuracies of the passkey retrieval at the document length of [4k, 8k, 16k, 32k, 64k, 128k, 256k], for each length we evaluate at 11 different depths of the document (from 0, 0.1, 0.2, ... to 1.0). In addition, for each location of the passkey (depth) in the document, we evaluate the model with five different passkeys to measure accuracy. As seen in Figure 8, the average passkey retrieval accuracy for Samba 1.7B almost reaches 100% in around 150 steps, while the accuracy for Mistral 1.6B remains low, demonstrating the extrapolation ability of the Samba architecture.

## D    ADDITIONAL ANALYSES

**How to train models with Sliding Window Attention (SWA)?**    Since SWA has linear complexity with respect to the sequence length, it seems alluring to trade off the batch size to have a longer training sequence length without substantially decreasing the training throughput. However, as shown in Table 9, when the sequence length is increased, the validation perplexity also increases in all context lengths due to smaller batch sizes (Varis & Bojar, 2021), and the optimal ratio of sequence length/window size observed is 2, resulting in a training length of 4096.

Table 9: Perplexity on SlimPajama of Llama-2-SWA 438M models trained on different context sizes and batch sizes. We fix the sliding window size as 2048 and the training tokens per step as 2M.

| Batch Size | Sequence Length | Training Speed ($\times 10^5$ tokens/s) | Validation Context Length | | | |
|---|---|---|---|---|---|---|
| | | | 2048 | 4096 | 8192 | 16384 |
| 1024 | 2048 (Full Attention) | 10.4 | **11.59** | 38.12 | 156.18 | 357.32 |
| 512 | 4096 | 9.88 | 11.87 | **11.16** | **10.69** | **10.61** |
| 256 | 8192 | 9.66 | 11.98 | 11.26 | 10.79 | 10.69 |
| 128 | 16384 | 9.48 | 12.37 | 11.63 | 11.12 | 11.02 |
| 64 | 32768 | 9.29 | 12.94 | 12.46 | 11.96 | 11.86 |

---

[2]https://huggingface.co/tiiuae/falcon-mamba-7b-instruct

Table 10: Perplexity on the SlimPajama validation set of different linear recurrent and sliding window attention models with Short Convolution (SC) modules added separately to query, key and value representations. For hybrid models, SC is applied only to linear attention layers. The training speed is measured on $8\times$A100 GPUs.

| Architecture | Size | Training Speed ($\times 10^5$ tokens/s) | Validation Context Length | | |
|---|---|---|---|---|---|
| | | | 4096 | 8192 | 16384 |
| Llama-2-SWA | 438M | 4.96 | 11.12 | 10.66 | 10.57 |
| + SC | 438M | 4.69 | 10.83 | 10.39 | 10.31 |
| Sliding GLA | 438M | 4.94 | 10.43 | 10.00 | 9.92 |
| + SC | 438M | 4.44 | 10.39 | 9.96 | 9.87 |
| Sliding RetNet | 446M | 4.32 | 10.38 | 9.96 | 9.87 |
| + SC | 446M | 3.80 | 10.25 | 9.82 | 9.74 |

**Fair comparison between Mamba and other linear recurrent models?** We can notice that the Short Convolution (SC) operator in Equation (1) is independent to the design of other parts of Mamba and can be applied to other linear recurrent models. As shown in Table 10, we explore the effect of SC on model performance through enhancing Llama-2-SWA, Sliding GLA, and Sliding RetNet with SC. Surprisingly, besides boosting the performance of RetNet, adding SC can also significantly improve the SWA's performance, while the effect on GLA is less prominent. We think this is because GLA already has the fine-grained decays at the channel level, so the depthwise convolution doesn't add much of the useful inductive bias for better modeling power. Notably, even with the SC enhancer, Sliding GLA and Sliding RetNet still fall short than the original Samba 421M's performance shown in Table 3. This further justifies our choice of using Mamba for hybridization. We also find that adding SC to both the SWA and the linear attention layers in hybrid models produces negative results, and we leave it as a future work to understand the surprising effectiveness of SC in language modeling.

## E    DETAILS OF ENTROPY MEASUREMENT

Given a causal attention probability matrix $A \in \mathbb{R}^{h \times n \times n}$, $A_{ijk} = 0\ \forall j < k$, with $h$ number of heads and a sequence length of $n$, and the generation length $0 < l < n$, we calculate the average attention entropy per decoding step as follows,

$$\mathcal{H}_a = -\frac{1}{l \cdot h} \sum_{i=1}^{h} \sum_{j=n-l+1}^{n} \sum_{k=1}^{n} A_{ijk} \log(A_{ijk}).$$

For the selective gate $\Delta \in \mathbb{R}^{n \times d_e}$ used by S6 in Equation (2) of the Mamba layers, we first normalize it to be in the simplex $[0, 1]^{n \times d_e}$, *i.e.*,

$$\Delta' = \frac{\Delta}{\sum_{i=1}^{n} \Delta_i} \ \in [0, 1]^{n \times d_e}.$$

The average selection entropy of S6 throughout the entire sequence is then calculated as

$$\mathcal{H}_s = -\frac{1}{d_e} \sum_{j=1}^{d_e} \sum_{i=1}^{n} \Delta'_{ij} \log(\Delta'_{ij}).$$

## F    DETAILS OF DOWNSTREAM LONG-CONTEXT EVALUATION

We use the GovReport (Huang et al., 2021) and the SQuALITY (Wang et al., 2022) datasets from the ZeroSCROLLS (Shaham et al., 2023) benchmark to evaluate models' long-context summarization capability in the real world. After tokenizing with the *Phi3-mini-4k* tokenizer, the average document length for the GovReport dataset is 11,533 tokens, with a median of 10,332, a minimum of 1,493, and a maximum of 40,592 tokens. For the SQuALITY dataset, the average sequence length is 7,974 tokens, with a median of 8,145, a minimum of 5,457, and a maximum of 10,757 tokens. For evaluation, we use greedy decoding for both tasks. A maximum generation length of 450 tokens is applied for GovReport and 600 for SQuALITY.

## G  IMPLEMENTATION DETAILS

Table 11: Detailed hyper-parameters of the baselines models trained on the Phi2 dataset with 230B tokens.

| Architecture | Llama-3 | Mistral | Mamba | Mamba-SWA-MLP | Mamba-MLP |
|---|---|---|---|---|---|
| Parameters | 1.6B | 1.6B | 1.8B | 1.6B | 1.9B |
| Batch size | 2048 | 2048 | 2048 | 2048 | 2048 |
| Learning rate | 0.0006 | 0.0006 | 0.0006 | 0.0006 | 0.0006 |
| Weight decay | 0.1 | 0.1 | 0.1 | 0.1 | 0.1 |
| Gradient clipping | 1.0 | 1.0 | 1.0 | 1.0 | 1.0 |
| Sequence length | 4096 | 4096 | 4096 | 4096 | 4096 |
| Sliding window size, $w$ | - | 2048 | - | 2048 | - |
| Number of layers, $N$ | 48 | 48 | 64 | 54 | 48 |
| Model width, $d_m$ | 2048 | 2048 | 2048 | 2048 | 2048 |
| MLP intermediate size, $d_p$ | 8196 | 8196 | - | 8196 | 8196 |
| Number of query heads | 32 | 32 | - | 32 | 32 |
| Number of KV heads | 4 | 4 | - | 4 | 4 |
| Number of Attention Layers | 24 | 24 | 0 | 18 | 0 |
| Number of Mamba Layers | 0 | 0 | 64 | 18 | 24 |
| Vocabulary size | 50304 | 50304 | 50304 | 50304 | 50304 |

For the GLA layer in the Sliding GLA architecture, we use the number of heads $d_m/384$, a key expansion ratio of 0.5, and a value expansion ratio of 1. For the RetNet layer we use a number of head that is half of the number of attention query heads, key expansion ratio of 1 and value expansion ratio of 2. The GLA and RetNet implementations are from the Flash Linear Attention (Yang & Zhang, 2024) repository[3] . We use the FlashAttention-based implementation for Self-Extend extrapolation[4]. The Mamba 432M model has a model width of 1024 and the Mamba 1.3B model has a model width of 2048. All models trained on SlimPajama have the same training configurations and the MLP intermediate size as Samba, unless otherwise specified. The training infrastructure on SlimPajama is based on a modified version of the TinyLlama codebase[5].

Table 12: Detailed hyper-parameters of the SAMBA models trained at different scales. We only show the optimization settings for the first training phase of the 3.8B model.

| Total Parameters | 421M | 1.3B | 1.7B | 3.8B |
|---|---|---|---|---|
| Dataset | SlimPajama | SlimPajama | Phi-2 | Phi-3 |
| Batch size | 512 | 512 | 2048 | 2048 |
| Learning rate | 0.0004 | 0.0004 | 0.0006 | 0.0006 |
| Total training tokens | 20B | 100B | 230B | 3.2T |
| Weight decay | 0.1 | 0.1 | 0.1 | 0.1 |
| Gradient clipping | 1.0 | 1.0 | 1.0 | 1.0 |
| Sequence length | 4096 | 4096 | 4096 | 4096 |
| Sliding window size, $w$ | 2048 | 2048 | 2048 | 2048 |
| Number of layers, $N$ | 24 | 36 | 48 | 64 |
| Model width, $d_m$ | 1536 | 2304 | 2048 | 2816 |
| MLP intermediate size, $d_p$ | 4096 | 6144 | 8196 | 9984 |
| Number of query heads | 12 | 18 | 32 | 11 |
| Number of key-value heads | 12 | 18 | 4 | 1 |
| Vocabulary size | 32000 | 32000 | 50304 | 32064 |

In the generation configurations for the downstream tasks, we use greedy decoding for GSM8K, and Nucleus Sampling (Holtzman et al., 2019) with a temperature of $\tau = 0.2$ and top-$p = 0.95$ for HumanEval. For MBPP and SQuAD, we set $\tau = 0.01$ and top-$p = 0.95$.

---

[3]https://github.com/sustcsonglin/flash-linear-attention
[4]https://github.com/datamllab/LongLM/blob/master/self_extend_patch/Llama.py
[5]https://github.com/jzhang38/TinyLlama

## H    LIMITATIONS & BROADER IMPACT

Although Samba demonstrates promising memory retrieval performance through instruction tuning, its pre-trained base model has retrieval performance similar to that of the SWA-based model, as shown in Figure 8. This opens up future direction on further improving the Samba's retrieval ability without compromising its efficiency and extrapolation ability. In addition, the hybridization strategy of Samba is not consistently better than other alternatives in all tasks. As shown in  Table 2, Mamba-SWA-MLP shows improved performance on tasks such as WinoGrande, SIQA, and GSM8K. This gives us the potential to invest in a more sophisticated approach to perform input-dependent dynamic combinations of SWA-based and SSM-based models (Ren et al., 2023). With the improved short-context performance and the long-term memorization ability of linear complexity LLMs such as Samba, cost-effective applications can be developed for personalized learning and automated tutoring. Samba can also be used for emotional accompaniment. The efficiency of the Samba architecture can save inference energy costs for models deployed on the edges, resulting in greener and more sustainable AI applications.

