# OpenReview forum: "Samba: Simple Hybrid State Space Models for Efficient Unlimited Context Language Modeling"
_ICLR.cc/2025/Conference — ICLR 2025 Poster_

### Official Review · Reviewer_MPi8 · 2024-10-26

**Soundness:** 3
**Presentation:** 3
**Contribution:** 3
**Rating:** 8
**Confidence:** 5

**Summary:**

The paper describe the hybrid Language model which combined Mamba layers and Sliding Windows Attention layers. Authors trained a number of models upto 3.8B size and showed that model performed well both on multiple standard LLM benchmarks and on some long context benchmarks (e.g. Passkey retrieval ). They also evaluated model on  Phonebook to show model limitation. Samba has better throughput comparing to regular Transformers both for  sequences with long prompt, and on generation long context.

**Strengths:**

Authors build a good LM which can both process long prompt and generate very long context  reliably. The simple idea - to combine mamba with SWA -  is very well executed, Solid experimental part, good ablation study. The paper is well written.

**Weaknesses:**

The main weakness of  Samba, is that similar to other hybrid models which use local attention it solves long context problem only partially. Experiments on Phonebooks show that the accuracy of the model rapidly goes down after 4K (Fig. 5).

I would also suggest to move the following observation from Appendix G  to main part of the paper, e.g. to Analysis or Conclusion section:
"Although Samba demonstrates promising memory retrieval performance through instruction tuning, its pre-trained base model has retrieval performance similar to that of the SWA-based model ... "

A few minor comments:
1. Inconsistent location of captions for figures and tables (sometimes above , sometimes below)\. Figure 3 is located inside text.
2. The implementation details: RoPE base is missing

**Questions:**

It would be interesting to see if Samba is able to handle
 a) more complex Needle-in-the-Haistack from RULER benchmark https://arxiv.org/abs/2404.06654
 b) complex multi--hop  reasoning tasks from HELMET benchmark https://arxiv.org/pdf/2410.02694

---

> ### Author Response · Authors · 2024-11-29
>
> We would like to thank the reviewer for the positive review of our work and the constructive feedback on our manuscript. In the following, we address the remaining concerns.
>
> > I would also suggest to move the following observation from Appendix G to main part of the paper, e.g. to Analysis or Conclusion section:
> "Although Samba demonstrates promising memory retrieval performance through instruction tuning, its pre-trained base model has retrieval performance similar to that of the SWA-based model ... "
>
> Thanks for pointing it out. We have added a similar description in L421-L423 of the revised **PDF**.
>
> > Inconsistent location of captions for figures and tables (sometimes above , sometimes below). Figure 3 is located inside text.
>
> Thanks. We have reorganized the location of captions and Figure 3 in the revised **PDF**.
>
> >The implementation details: RoPE base is missing
>
> The RoPE base is 10,000 and we added it in L168 of the revised **PDF**.
>
> >It would be interesting to see if Samba is able to handle a) more complex Needle-in-the-Haistack from RULER benchmark https://arxiv.org/abs/2404.06654 b) complex multi--hop reasoning tasks from HELMET benchmark https://arxiv.org/pdf/2410.02694
>
> Thanks for pointing them out! We plan to evaluate Samba on them in the near future.

---

### Official Review · Reviewer_4dj2 · 2024-11-02

**Soundness:** 4
**Presentation:** 4
**Contribution:** 3
**Rating:** 6
**Confidence:** 4

**Summary:**

Authors investigate the utility of attention-state-space hybrids over attention-only models (e.g. sliding window attention (SWA)) and state-space-only models (e.g. Mamba) by training language models in the 400M-4B parameter range. They propose a model block Mamba->MLP->SWA->MLP and perform ablation studies in a language modeling (LM) setup. Pretrained models are evaluated 0-shot on 15 established downstream benchmarks.

**Experiments**

1. Authors perform an ablation study on the Phi-2 training corpus (230B tokens) and report modest improvements over the Mamba->SWA->MLP baseline. On reading comprehension tasks such as SQuAD, the addition of SWA to Mamba leads to significant improvement over the Mamba-only baseline.

2. Authors explore replacing Mamba in the hybrid block with other efficient alternatives such as Gated Linear Attention and RetNet. Models are trained on SlimPajama corpus with 4k length and evaluated on lengths up to 16k, with the Mamba-based block reporting the lowest perplexity.

3. Authors study the length extrapolation abilities of models trained with 4k length. On the ProofPile test set, the hybrid block shows improved and slowly decreasing perplexity with increasing context length, and is lower than that of the SWA-only baseline.

4. Authors investigate the model's performance on memory-intensive tasks. After finetuning for 500 steps with 4k input length, the hybrid block achieves perfect performance on PassKey Retrieval for lengths up to 256k. On the PhoneBook task of retrieving phone numbers from name-number pairs, models finetuned for 100 steps with 4k input length show superior performance compared to full attention when evaluated on lengths up to 8k.

5. Analysis of attention scores reveals that middle layers have lower entropy compared to top/bottom layers, suggesting specialization in precise information retrieval.

6. The hybrid model trained on Phi-3's 3.2T token corpus matches parameter-matched Phi-3's performance on downstream tasks.

**Strengths:**

1. The investigation of SSM-attention hybrids is well motivated and timely. The empirical findings and model performance should interest a broad audience. The trained checkpoints would be a useful resource for the research community.

2. Experiments are comprehensive, showing significant improvements in length extrapolation tasks and efficiency on long inputs.

**Weaknesses:**

1. Limited novelty - adding an MLP in the model block is an incremental contribution. The performance gap between Mamba->MLP->SWA->MLP and Mamba->SWA->MLP is modest.

2. The models' strong performance relies on proprietary datasets (Phi-2 and Phi-3), making fair comparisons difficult.

3. The relationship with previous SSM+SWA hybrid works (Griffin/Hawk) needs clearer positioning.

**Questions:**

1. In models with SSM layers, was RoPE still necessary in SWA? (This doesn't affect my evaluation)

2. Are the Phi-2 and Phi-3 training sets open-sourced? Reproducibility is challenging without access to these datasets.

3. Please add Mamba->SWA->MLP baseline to Table 3.

4. Consider including Mistral results on PhoneBook.

---

> ### Author Response · Authors · 2024-11-29
>
> We would like to thank the reviewer for the positive review of our work and the constructive feedback on our manuscript. In the following, we address the remaining concerns to hopefully motivate a clear acceptance score.
>
> **Limited Novelty & Modest Performance gap between Samba and Mamba-SWA-MLP:** We want to first point out that both Samba and Mamba-SWA-MLP are first proposed by us. We are the first to show that interleaving Mamba with both SWA and MLP can substantially outperform modern Transformers (and Mamba) on short-context tasks at a scale up to 3.8B parameters, while achieving comparable training speed and better length extrapolation ability under the perplexity metrics. For similar performance between Samba and Mamba-SWA-MLP, we want to point out the Samba has faster training speed than Mamba-SWA-MLP, as shown in *Updated Perplexity and Downstream Results* in **General Response**,  and it saves more KV/SSM state cache because of fewer Mamba and SWA layers if we tie the total number of parameters to be the same. We have added these explanation in L322-325 in the revised **PDF**.
>
> **Proprietary Datasets:** To improve the reproducibility, we also evaluate our 1.3B models pre-trained on SlimPajama on common downstream benchmarks. Samba is still the best architecture with respect to both efficiency and performance and outperforms LLama-2 by a large margin. Please refer to *Updated Perplexity and Downstream Results* in **General Response** for more details.
>
> **Relationship with Griffin:** We actually have positioned our work in the context of Griffin. Please refer to L983-988 in the revised **PDF**.
>
> ---
>
> **Q1:** *In models with SSM layers, was RoPE still necessary in SWA?*
>
> **A:** We trained Samba-NoPE at both 421M and 1.3B scales on SlimPajama. Interestingly, even though we use SWA in
> Samba architecture, Samba-NoPE still has exploded perplexities beyond its training length. However, if we evaluate them within the training sequence length, they do have similar downstream performance. Please refer to the *Updated Perplexity and Downstream Results* section in **General Response** for more details.
>
> **Q2:** *Are the Phi-2 and Phi-3 training sets open-sourced?*
>
> **A:**  Unfortunately, they will not be open-sourced in the short-term, but Samba also works great on SlimPajama.
>
> **Q3:** *Please add Mamba->SWA->MLP baseline to Table 3.*
>
> **A:** Added it. Please refer to Table 3 in the revised **PDF**.
>
> **Q4:** *Consider including Mistral results on PhoneBook.*
>
> **A:** Thanks for the suggestion. Added it! Please refer to the *Updated Phonebook Results* section in **General Response** for more details.

---

> ### Author Response · Authors · 2024-12-01
> **Reminder on paper discussion**
>
> Dear reviewer 4dj2,
>
> As we approach the end of the discussion period, we would greatly appreciate your input on the paper. We hope our responses and additional results address your concerns and welcome any further questions or suggestions.
>
> Thank you for your time and effort.

---

### Official Review · Reviewer_A57R · 2024-11-08

**Soundness:** 3
**Presentation:** 3
**Contribution:** 3
**Rating:** 6
**Confidence:** 4

**Summary:**

This paper proposes a hybrid architecture mixing Mamba layer with Sliding Window Attention.

**Strengths:**

- The proposed architecture shows great results on common benchmark.
- It seems also to be better at long contexts handling.
- Shows that hybrid models can be competitive alternatives to transformers.

**Weaknesses:**

- The main contribution of combining attention with other linear mechanisms is not novel, and, as noted in the paper, a lot of alternatives exist.
- A comprehensive benchmarking against existing alternatives is lacking. Comparisons are only made to their proposed variants and Sliding Window Attention in fair setups. A thorough comparison with other models listed in Appendix A (such as MEGA, adapted to Mamba) would strengthen the findings. Additionally, selected architectures are evaluated on a very small scale and only measured by perplexity. While some models achieve lower perplexity, this alone may not suffice to establish superiority (e.g., in H3 by Dao et al., 2022b, lower perplexity is reported against transformer baselines).
- Results on common benchmarks are somewhat misleading. The paper aims to showcase the architecture’s strengths, yet comparisons are often made against models trained on different data distributions, which weakens the robustness of the conclusions.
- Conclusions on long-context handling remain vague, although this should be a key advantage over transformers. It would be helpful to include dataset statistics (average, median, min, max lengths) to clarify context length relevance.
- The only substantial long-context experiment, the summarization task, should be included in the main paper, with clearer discussion and analysis.
- Section 4, “Analysis,” could benefit from clearer motivation. Some explored dimensions may appear intuitive (e.g., l. 444, where SWA is shown to outperform full attention on larger sequence lengths than those used in training), which might limit the novelty of the findings. Other questions seems a bit unrelated to the paper topics (see Questions).
- Length extrapolation, a key aspect of the paper, is barely motivated or discussed in relation to prior work.
- The paper overall feels somewhat unstructured and difficult to follow. Tables present different baselines inconsistently, and messages regarding architectural advantages are interleaved with comments on training data quality (l. 229). The evaluation setup lacks consistency (performance is sometimes assessed on real benchmarks, other times by perplexity), and the rationale behind baseline choices or research questions is insufficiently explained.

**Questions:**

- In the original Mamba paper, they already experiment with mixed Mamba/Attention. Can you contextualize the your contribution with respect to this original architecture?

- What is the evaluation setup for the summarization in Appendix B? Also, please bold the best performance on SQUALITY.

- I’m having difficulty identifying the key factors that contribute to the optimal architectural design. What elements are beneficial here? Is it the MLP, Mamba, or both, and to what extent? I’d be interested in seeing results for a SWA > MLP > SWA > MLP architecture as an example (where some SWA layers are substituted with MLP).

- In the length extrapolation experiments, can you clarify whether the models can extrapolate up to 256k tokens in Passkey Retrieval, but struggle with generalization in Phonebook performance?

- In several sections, you mention that Full Attention performance deteriorates beyond a specific context size. This seems logical, couldn't these attention be converted to sliding attention especially l. 444?

- In addition to the training curves shown in Fig. 7 of Appendix D, could you also provide the validation curves?

- In Section 4, I don’t fully understand the relevance of the question: "How to train models with Sliding Window Attention (SWA)?"

- This question might benefit from rephrasing, since your response is exploratory rather than definitive, focusing on the distribution of attention rather than giving a clear answer.

- Line 051: The phrase "limited context extrapolation ability in perplexity metrics" is unclear. Are you suggesting that the extrapolation metrics increase perplexity when the models exceed their initial training context size?

- Line 164 implies that SWA was specifically designed to address Mamba’s limitations, this is misleading.

---

> ### Author Response · Authors · 2024-11-29
>
> We would like to thank the reviewer for the constructive feedback on our manuscript. In the following, we address the remaining concerns to hopefully encourage a positive evaluation.
>
> **Missing Baselines:** Thanks for the suggestion of adding Mamba-adapted Mega as a baseline architecture. Please refer to the *Updated Perplexity and Downstream Results* section in the **General Response** for more details.
>
> **Lack of Downstream Evaluation:** Thanks for pointing it out. In addition to our originally presented evaluation results of models trained on Phi2 and Phi3, we also added the downstream evaluation results for our 1.3B models trained on SlimPajama. Samba is still the best model with respect to both efficiency and performance. Please refer to **General Response** for more details.
>
> **Misleading Benchmarks:** We want to point out that the captions of the tables at 3.8B scales have emphasized that only Phi3 and Samba are fairly comparable, and other open source LLMs are only for reference. Apart from the fair comparisons made at the 1.7B and 3.8B scale for common benchmark, we also add rigorous downstream benchmarking for models at a scale of 1.3B for improved reproducibility. We agree that it is kind of wasting the valuable space in the main part of our paper to include less informative comparisons, and we switch in the table with only the post-trained Samba and Phi3 for more focus on architectural differences.  Please refer to Table 1 in the revised **PDF** for more details.
>
> **Missing Long-context Data statistics:** Thanks for the suggestion. We have added them in the Appendix F of our revised **PDF**.
>
> **Analysis Motivation:** The analysis "How to train models with Sliding Window Attention (SWA)?" is to support our choices of 2K SWA when training on 4K sequence length. The ratio of 0.5 between window size and the input sequence length is crucial because we show that 2K SWA has both comparable perplexity and faster training speed than full attention at 4K input sequence length. This motivates us to use 2K SWA to replace Mamba in Mamba-MLP instead of full attention, which results in our final Samba architecture. We agree that it may appear less relevant to other analyses and have moved it to Appendix D.
>
> **Related Work for Length Extrapolation:** Thanks for the suggestion. We added it and please refer to *Extended Related Works* in **General Response** for more details.
>
> **Unstructured Paper:** Thanks for the comment. We hope the revised paper is now more structured.
>
> ---
>
> **Q1:** *...contextualize...contribution with respect to this original architecture?*
>
> **A:** Yes, please refer to *Extended Related Works* in **General Response**.
>
> **Q2:** *...evaluation setup for the summarization in Appendix B? Also, please bold the best performance...*
>
> **A:** Please refer to the Appendix F of our revised **PDF**. Thanks, we have bolded it.
>
> **Q3:** *...What elements are beneficial here? Is it the MLP, Mamba, or both, and to what extent?...*
>
> **A:** We added the MLP->MLP->SWA->MLP (named MLP2-SWA-MLP) in *Updated Perplexity and Downstream Results* of the **General Response**. The results suggest that both MLP and Mamba are beneficial.
>
> **Q4:** *...struggle with generalization in Phonebook performance?*
>
> **A:** Thanks for the insightful question! Our current observation is that the Samba 1.7B can not transfer its number recall ability from Passkey to Phonebook. Please refer to the *Updated Phonebook Results* in **General Response** for more details.
>
> **Q5:** *...couldn't these attention be converted to sliding attention...*
>
> **A:** Yes, full attention can be converted to sliding window attention with attention sink (or lambda attention) as in StreamingLLM [1] and LM-infinite [2]. However, after the conversion, they cannot show improved perplexity beyond its training sequence length (please refer to Figure 3 in [1] ). In contrast, we prove that if we pretrain SWA from scratch, SWA is naturally extrapolatable on perplexity.
>
> **Q6:** *...could you also provide the validation curves?*
>
> **A:** Yes, it is provided in the Figure 8 of Appendix C in the revised **PDF**, whose validation accuracy is calculated on 2048 examples which are generated on the fly.
>
> **Q7:** *...I don’t fully understand the relevance of the question*
>
> **A:** Please refer to our response in the **Analysis Motivation** section above.
>
> **Q8:** *This question might benefit from rephrasing...*
>
> **A:** We have rephrase it in L480 in the revised **PDF**.
>
> **Q9:** *The phrase "limited context extrapolation ability in perplexity metrics" is unclear....*
>
> **A:**  Thanks for pointing it out. We have rephased it and please refer to L49-L51 in the revised **PDF**.
>
> **Q10:** *Line 164...is misleading.*
>
> **A:** Thanks, we have rephrased it in L164-L165 in **PDF**.
>
> ---
>
> [1] Efficient Streaming Language Models with Attention Sinks (ICLR 2023)
>
> [2] LM-Infinite: Zero-Shot Extreme Length Generalization for Large Language Models (NAACL 2024)

---

> ### Author Response · Authors · 2024-12-01
> **Reminder on paper discussion**
>
> Dear reviewer A57R,
>
> As we approach the end of the discussion period, we would greatly appreciate your input on the paper. We hope our responses and additional results address your concerns and welcome any further questions or suggestions.
>
> Thank you for your time and effort.

---

> ### Comment · Reviewer_A57R · 2024-12-02
> **Response to authors**
>
> Thank you to the authors for their response and consideration of my feedback. The paper is now clearer, and the contribution is more effectively highlighted. I have updated my scores accordingly.

---

### Author Response · Authors · 2024-11-29
**General Response**

We thank the reviewers for their constructive feedback. We have revised the **PDF** and highlighted our revisions in purple. Major revisions are summarized in the following sections.

## Updated Perplexity and Downstream Results

We train the following four alternative architectures with both around 400M and 1.3B parameters on the SlimPajama dataset to provide more comprehensive and reproducible benchmarking and ablation studies:

- Mega-S6: We replace all MD-EMA modules in the Mega architecture [1] with the ShortConv+S6 combinations from Mamba to adapt Mega to the modern Mamba architecture. Rotary position embedding, RMSNorm and Softmax attention are also adopted. We set the intermediate dimension of the Mega-S6 layer to be $d_m$ so that it has a roughly $5d_m^2$ number of parameters. This represents a classical baseline that conducts intra-layer SSM-Attention hybridization sequentially.

- Mamba-SWA-MLP: The Mamba->SWA->MLP architecture as in Figure 1 of our Paper.

- MLP2-SWA-MLP: We replace all Mamba layers in the Samba architecture to SwiGLU layers with $6d_m^2$ number of parameters.

- Samba-NoPE: We remove the rotary relative position embedding in Samba.

In the Table 3 of the revised **PDF** (also partially shown below), we can first notice that Mega-S6 shows significantly worse performances than other inter-layer hybridization strategies. Comparing Mamba-SWA-MLP with Samba, we can see that Samba has both slightly better perplexity scores and higher training throughput. Mamba-SWA-MLP trades off the MLP layers with more I/O intensive Mamba and Attention layers, leading to slower training speed. This also indicates that Mamba-SWA-MLP will have slower decoding speed than Samba due to larger total cache size resulting from more SSMs and Attention layers. We can observe that replacing Mamba with MLP in speeds up the training but harms perplexity significantly, indicating the importance of the Mamba layers in the Samba architecture. Interestingly, although we use SWA in Samba, Samba-NoPE has exploded perplexity beyond its training length without RoPE.

| **Architecture**    | **Size** | **Layers** | **Training Speed** | **Val. @ 4k** | **Val. @ 8k** |  **Val. @ 16k** |
|---|--|--|--|-|-|-|
| *20B training tokens on 8×A100 GPUs* |   |   |  |  |  |  |
| Mega-S6  | 422M | 24 | 3.26  | 12.63 | 12.25| 12.25 |
| Mamba-SWA-MLP  | 400M    | 24   | 4.21  | 10.07  | 9.67 | 9.59  |
| MLP2-SWA-MLP    | 417M     | 24  | **5.08**  | 10.95  | 10.50| 10.41 |
| Samba-NoPE  | 421M     | 24     | 4.48     | 10.11 | 28.97| 314.78|
| Samba  | 421M     | 24      | 4.46      | **10.06**  | **9.65**| **9.57** |
| *100B training tokens on 64×H100 GPUs* |     |   |    |    |  |   |
| Mega-S6      | 1.3B     | 36     | 17.9     | 9.01 | 8.81 | 8.68  |
| Mamba-SWA-MLP | 1.3B     | 36    | 23.5   | 7.37  | 7.16 | 7.00  |
| MLP2-SWA-MLP    | 1.3B     | 36     | **26.6**   | 7.81     | 7.58 | 7.42  |
| Samba-NoPE      | 1.3B     | 36     | 25.2  | 7.33    | 20.40| 326.17|
| Samba | 1.3B     | 36    | 25.2    | **7.32**  | **7.11**| **6.96** |

In the Table 4 of the revised **PDF** (also shown below), we evaluate all our ~1.3B scale models pre-trained with 100B tokens from SlimPajama on five typical commonsense reasoning tasks to understand the effect of architecture designs on downstream performances. We can see that Samba has the best average accuracy, outperforming the LLaMA 2 architectures by a large margin. Similar to our perplexity evaluation, Samba and Samba-NoPE have similar average accuracies, whereas Mamba-SWA-MLP falls slightly behind. We observe that different architectures excel at different tasks. Mamba-SWA-MLP performs best on ARC-Easy, while Samba and Samba-NoPE achieve superior results on LAMBADA. Hybrid models based on Mamba generally outperform hybrid linear attention models and pure softmax-attention models on HellaSwag.



| **Architecture**      | **Size** | **ARC-Easy (acc ↑)** | **HellaSwag (acc_norm ↑)** | **WinoGrande (acc ↑)** | **PIQA (acc ↑)** | **LAMBADA (acc ↑)** | **Avg.** |
|--|--|--|--|--|--|-|--|
| LLaMA-2     | 1.3B    | 55.09    | 52.32  | 53.35    | 71.11   | 48.52    | 56.08    |
| LLaMA-2-SWA  | 1.3B    | 56.65  | 52.59   | 54.93      | 71.60  | 47.56   | 56.67    |
| Sliding GLA      | 1.2B    | 56.94  | 52.52  | **56.75**  | 71.38      | 48.17    | 57.15    |
| Sliding RetNet   | 1.4B    | 57.66   | 52.64    | **56.75**     | 71.33    | 48.34    | 57.34    |
| Mega-S6 | 1.3B    | 50.63   | 41.91   | 52.96    | 68.17   | 37.88   | 50.31    |
| Mamba   | 1.3B    | 58.08    | **54.93**  | 53.99  | 71.98   | 45.97   | 56.99    |
| Mamba-SWA-MLP   | 1.3B    | **59.64**    | 54.50  | 55.25  | **72.42**  | 49.12 | 58.19    |
| MLP2-SWA-MLP | 1.3B    | 55.18   | 50.32  | 52.80  | 70.67  | 48.11      | 55.42    |
| Samba-NoPE | 1.3B    | *58.38*  | 54.62   | 56.51  | 72.03    | *51.08*      | *58.52*  |
| Samba  | 1.3B    | 58.21 | *54.73*  | 55.72   | *72.36*  | **51.68**  | **58.54** |

---

> ### Author Response · Authors · 2024-11-29
> **General Response (continued)**
>
> ## Updated Phonebook Results
>
> Since both Passkey Retrieval and Phonebook require models to remember numbers in a long context document, it is interesting to investigate if a model instruction-tuned on one task can transfer its ability to the other task in zero-shot. We directly evaluate the Passkey Retrieval finetuned Samba 1.7B and Mistral 1.6B models (named Samba 1.7B PK-FT and Mistral 1.6B PK-FT respectively) on the Phonebook task. As shown in Figure 4 of the revised **PDF**, Samba 1.7B has slightly better retrieval accuracy than Mistral 1.6B, but both models cannot generalize their number recall ability beyond its sliding window size. We leave it for future work to further explore the transferability of long-context capabilities in linear complexity models.
>
> ## Extended Related Works
>
> We also added the following paragraphs in Appendix A in the revised **PDF** to better contextualize our work in the literature of hybrid recurrent models and length extrapolation.
>
> The original Mamba paper also explores hybridizing pure Mamba models with full attention or MLP layers, but it does not consider the wall-time efficiency of these hybridization and only achieves marginally better performance than the pure Mamba model. In contrast, we are the **first** to show that interleaving Mamba with both SWA and MLP can substantially outperform modern Transformers (and Mamba) at a scale up to 3.8B parameters, while achieving comparable training speed and better length extrapolation ability under the perplexity metrics.
>
> Many previous works have focused on extending the context length of pretrained Transformers to improve their performance on long-context tasks. Methods such as LM-Infinite [2], StreamingLLM [3] and LongLoRA [4] achieve linear complexity for length extrapolation, but they only stabilize the perplexity beyond the training sequence length. However, we show that if we pre-train Transformers with Sliding Window Attention from scratch, it can naturally have improved perplexity beyond the training sequence length. Other approaches, including LLaMA-2-Long [5], LongLLaMA [6], PI [7], LongRoPE [8] and Self-Extend [9], attempt to extend the full attention through modifying position embedding or continual training strategies, but they typically retain quadratic complexity in the attention mechanism with additional computation or memory I/O overhead, therefore they do not scale well to very long sequences. Although these methods achieve an improved perplexity on a sequence length that is multiple times longer than the training sequence length, their perplexity still explodes if the sequence is extremely long. Our method achieves both linear complexity and superior extrapolation performance compared to zero-shot length extrapolation methods, such as Self-Extend, under the perplexity metric. However, we acknowledge that, in terms of zero-shot retrieval performance, our method still lags behind these approaches. This underscores a trade-off between perplexity and retrieval performance in length extrapolation, which we plan to explore and address in future work.
>
> ---
> [1] Mega: Moving Average Equipped Gated Attention (ICLR 2023)
>
> [2] LM-Infinite: Zero-Shot Extreme Length Generalization for Large Language Models (NAACL 2024)
>
> [3] Efficient Streaming Language Models with Attention Sinks (ICLR 2023)
>
> [4] LongLoRA: Efficient Fine-tuning of Long-Context Large Language Models (ICLR 2023)
>
> [5] Effective long-context scaling of foundation models (NAACL 2024)
>
> [6] ​​Focused transformer: Contrastive training for context scaling (NeurIPS 2023)
>
> [7] Extending context window of large language models via positional interpolation (arXiv 2023)
>
> [8] LongRoPE: Extending LLM Context Window Beyond 2 Million Tokens (ICML 2024)
>
> [9] LLM Maybe LongLM: Self-Extend LLM Context Window Without Tuning (ICML 2024)

---

### Meta-Review · Area_Chair_io5b · 2024-12-24

**Metareview:**

The paper explores the design of state space models (SSMs) that integrate components from Mamba, sliding window attention, and MLPs. The objective is to develop efficient architectures for language modeling that achieve linear time complexity, effectively handle long contexts, and support considerable-length extrapolation. This is primarily an experimental study demonstrating high-level performance across a range of benchmarks, surpassing state-of-the-art (SOTA) SSMs and delivering substantial speedups compared to transformer baselines. Additionally, the authors provide a comprehensive analysis of the model, justifying their design choices.

The reviewers acknowledge that while the technical novelty may be limited, the proposed model delivers notable improvements over SOTA SSMs and transformers. It also offers valuable experimental insights that represent a significant contribution to the community. During the rebuttal phase, the authors included new experiments addressing the reviewers’ initial requests. I recommend acceptance.

**Additional Comments On Reviewer Discussion:**

After rebuttal all the reviews are positive.

---

### Decision · Program_Chairs · 2025-01-22

Accept (Poster)